# Exploring the Capability of Natural Flood Management Approaches in Groundwater-Dominated Chalk Streams

**Imogen Barnsley** [1,*], **Rebecca Spake** [1], **Justin Sheffield** [1], **Julian Leyland** [1], **Tim Sykes** [2] **and David Sear** [1]

[1] School of Geography and Environmental Science, The University of Southampton, Southampton SO17 1BJ, UK; R.Spake@soton.ac.uk (R.S.); Justin.Sheffield@soton.ac.uk (J.S.); J.Leyland@soton.ac.uk (J.L.); d.sear@soton.ac.uk (D.S.)

[2] Romsey Office, The Environment Agency, Romsey SO51 8DU, UK; tim.sykes@environment-agency.gov.uk

[*] Correspondence: i.c.smith@soton.ac.uk

**Abstract:** This study aims to address the gap in the Natural Flood Management (NFM) evidence base concerning its implementation potential in groundwater-dominated catchments. We generated a typology of 198 chalk catchments using redundancy analysis and hierarchical clustering. Three catchment typologies were identified: (1) large catchments, (2) headwater catchments with permeable soils, and (3) catchments with impermeable soils and surfaces (urban and suburban land uses). The literature suggests that natural flood management application is most effective for catchments <20 km$^2$, reducing the likelihood of significant flood mitigation in large catchments. The relatively lower proportion of surface runoff and higher recharge in permeable catchments diminishes natural flood management's likely efficacy. Impermeable catchments are most suited to natural flood management due to a wide variety of flow pathways, making the full suite of natural flood management interventions applicable. Detailed groundwater flood maps and hydrological models are required to identify catchments where NFM can be used in a targeted manner to de-synchronise sub-catchment flood waves or to intercept runoff generated via groundwater emergence. Whilst our analysis suggests that most chalk groundwater-dominated catchments in this sample are unlikely to benefit from significant flood reductions due to natural flood management, the positive impact on ecosystem services and biodiversity makes it an attractive proposition.

**Keywords:** natural flood management; chalk streams; redundancy analysis; hierarchical clustering; surface runoff; base flow; transmissivity; recharge; groundwater flooding

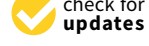



## 1. Introduction

Natural Flood Management (NFM) represents a paradigm of management strategies that aim to improve a catchment's resilience to prolonged and/or heavy rainfall by restoring, enhancing, or altering a catchment's natural hydrological and morphological characteristics [1]. Strategies aim to increase interception and infiltration by reducing rapid runoff generation, increasing catchment water storage, and slowing overland channel flows [2,3]. Flood peak reductions of up to 30–40% have been attributed to NFM methods, such as storage pond networks, tree shelter belts, channel realignment, leaky dams, and winter cover crops in rural and urban settings [2,4–8]. Moreover, NFM can offer co-benefits, such as improvements to water quality, reductions in soil degradation, and enhanced biodiversity [1,9,10]. Consequently, the demand for NFM implementation has increased due to its perception as a relatively low cost, low maintenance flood mitigation solution that protects and maintains hydrological and biological function of the rivers in which it is implemented [1,11,12]. Despite this growing demand for NFM interventions, the NFM evidence base consistently cites groundwater-dominated river systems, like chalk streams, as a gap in the knowledge due to hydrological differences, meaning the current evidence may not be directly applicable [11,13,14].

A small number of chalk streams can be found in Northern France and Russia, but the 224 chalk streams found in the UK account for the majority of this river type found globally [15]. Catchments underlain with chalk bedrock are predominantly groundwater dominated [16]. This means that large quantities of water are stored and transmitted through the chalk bedrock before being released into river channels as baseflow, generating a stable annual river regime with a lack of spate conditions and relatively small variations between high and low flows [17]. It can take months for changes in rainfall inputs to manifest as changes in the river regime [18], giving chalk streams a strong seasonal flow oscillation in conjunction with annual rainfall patterns in the UK. Discharge rises slowly over the winter, peaking in late April after heavier winter rainfall, and gradually recedes over the summer, with lowest flows in autumn after generally drier summers. Groundwater flooding occurs when rainfall recharge causes the groundwater level to rise, in turn increasing groundwater inputs into river systems and causing groundwater emergence in topographic low points, winterbournes, and activating springs [19,20].

NFM functions by enhancing a catchments' natural ability to absorb shocks from storms by storing water in the catchment and then slowly releasing it [21]. A successful NFM scheme reduces discharge at at-risk locations by extending a flood duration and reducing the peak discharge (and water level) at a point at any given time [13,22]. This is achieved by increasing interception and infiltration, slowing overland channel flows, and manipulating channel and catchment surface roughness [2,3]. NFM is a process-based approach, meaning that each NFM scheme is designed by matching flow pathways, landscape features, and sources of flood waters that contribute to peak flows to specific NFM interventions that tackle those issues. Because of this, NFM interventions can be categorised according to the processes that they manipulate: (1) the reduction of rapid runoff generation, (2) increasing catchment water storage, and (3) strategies to reduce the conveyance of water downstream [11,13,14]. This is an important distinction because it allows specific sources of flood waters to be linked to specific solutions in the process of NFM design. For example, many arable fields suffer from sediment loss due to excess rapid overland flow. In this case, NFM interventions would focus on reducing runoff inputs to the river channel by intercepting and storing water. Sediment can be trapped and stored and surface roughness increased (slowing the flow of surface runoff) by using winter cover crops or across-slope tillage. The choice of method would be dictated by the specific soil properties of the site as well as management preferences. As such, NFM schemes are often tailor-made to each application scenario, and it is important that the features and water-transfer processes of the catchment where implementation is proposed are first established.

In contrast, the vast majority of the NFM evidence base is founded on research conducted in surface water-dominated catchments where floods are caused by the convergence of multiple surface-runoff inputs to the river channel [11,13,23]. As a result, the application of NFM in groundwater-dominated catchments has been highlighted as a one of the key knowledge gaps in the NFM evidence base, in acknowledgement that the flow pathways and processes present in groundwater-dominated catchments are significantly different to those found in catchments with fluvial floods. This can be demonstrated by the difference in Base Flow Index (BFI) values for the River Lambourne (a typical chalk stream) and the Belford Burn catchment in Northumberland, which has been used as an NFM experimentation catchment for many years. BFI measures the proportion of total channel flow contributed by groundwater sources. Belford Burn has a BFI of 0.313 [24], and the River Lambourne has a BFI of 0.98 [25]. Therefore, groundwater catchments typically transfer a large proportion of water below the ground surface. Of the 12 main NFM techniques summarised by Lane (2017) [13], eight of them rely on managing surface water. However, if most of the water transfer throughout the catchment occurs below the ground surface, these measures may have limited effect (Figure 1). Because NFM interventions are tailored to fit the specific sources and flood water, it follows that NFM schemes in groundwater catchments will focus on different combinations of interventions than those found

in surface-runoff-dominated catchments (Figure 1) depending on the key properties of chalk catchments. It is therefore important to identify the morphological and hydrological features that affect groundwater recharge and the production of overland flows (increasing the probability of channel bank exceedance) to guide future NFM strategies in catchments dominated by groundwater processes.

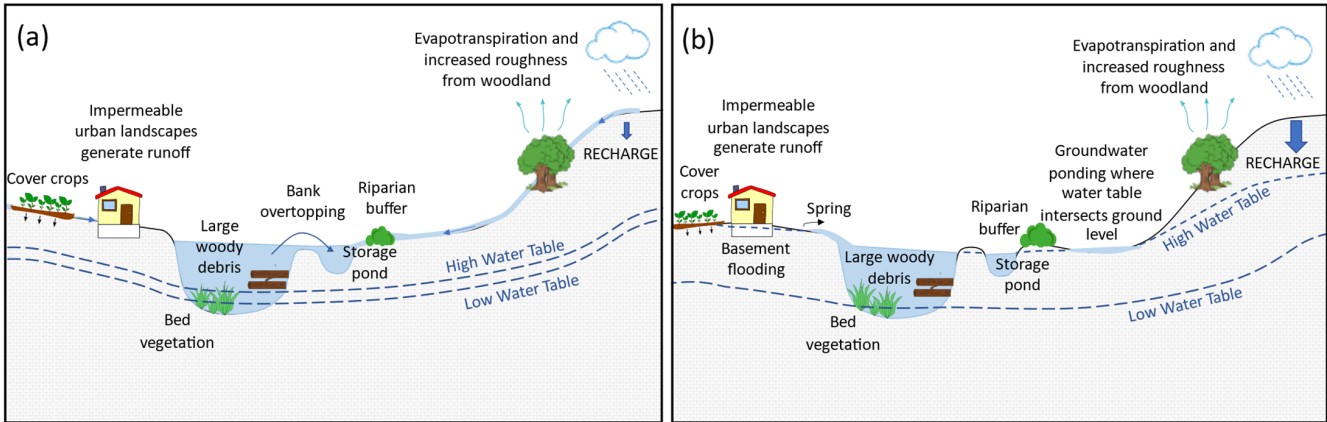

**Figure 1.** The processes of flooding in impermeable upland catchments (**a**) and in permeable chalk catchments (**b**) along with common NFM interventions to demonstrate how groundwater dominance in flood production may impact the efficacy of NFM.

This study is intended as a screening process for NFM in groundwater catchments by grouping catchments according to hydrological properties and matching them to NFM interventions that specifically tackle these flow pathways. To do this, we quantify the relationships between hydrological variability and key morphological characteristics that are amenable to NFM for 198 catchments with chalk bedrock in the Southeast of England. We use these results to classify the catchments and infer flow pathways and to make suggestions for the most appropriate NFM strategies for these river basins. Because NFM schemes must be designed on a case-by-case basis for the greatest effectiveness, this is intended as a broad-scale screening process to narrow down options and not as a comprehensive guide for choosing NFM interventions in all chalk stream catchments.

## 2. Materials and Methods

### 2.1. Study Area and Catchment Selection

Chalk groundwater-dominated catchments were first identified using topographic and geological datasets. Topographic catchment boundaries were provided by the Centre for Ecology and Hydrology National River Flow Archive (NRFA) [26]. A bedrock map (BGS 625k) from the British Geological Survey [27] was used to identify catchments that were underlain by chalk bedrock. To reduce uncertainty due to differing groundwater transfer processes [28] and because of the global rarity of chalk stream river systems, we focused on chalk groundwater-dominated river systems, excluding limestones or Permo-Triassic sandstones. We selected catchments with at least $\geq$70% of the catchment within the chalk bedrock and with gauging stations within 5 km downstream of the chalk bedrock (determined via the buffer tool in ArcGIS and visual inspection). Using these criteria, a total of 198 catchments were available for analysis, located predominantly in the Southeast of England (Figure 2).

### 2.2. Data Analysis

To relate hydrological variability to catchment morphological characteristics, we compiled and analysed the covariation among four key hydrological variables and twenty-one variables quantifying the physical catchment properties. Details and data sources are found in Table 1.

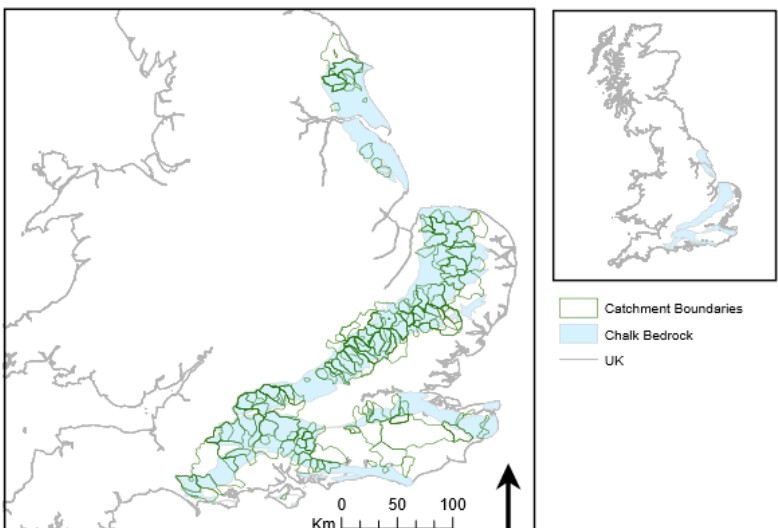

**Figure 2.** Location of the 198 catchments included in this study underlain by chalk bedrock. Catchment boundaries are from the National River Flow Archive, and the bedrock map is from the British Geological Survey.

### 2.2.1. Hydrological Variables

All hydrological variables were derived from the Centre for Ecology and Hydrology UK National River Flow Archive (NRFA) [26]. Average discharge ($Q_{mean}$) and maximum recorded discharge ($Q_{max}$) are among the most basic hydrological metrics used to characterise the hydrological regime. The base flow index (BFI) [25] measures the proportion of streamflow from groundwater contributions. This is relevant to NFM implementation because high BFI values are related to stable flows and limited surface-runoff generation and contributions to streamflow. Conversely, the Richard–Baker flashiness index (RBFI) [29] measures the sensitivity of a streamflow to rainfall inputs. Therefore, high RBFI values indicate significant sources of runoff resulting in spate conditions. As mentioned previously, understanding the processes by which water is transferred throughout the catchments is essential to NFM design. Chalk streams are often characterised and defined by features such as long times to peak, high base flow domination, and a lack of spate conditions [15]. However, large ranges in BFI and RBFI values within the chalk stream sample demonstrate that there is major spatial variation in chalk stream properties. These metrics will help characterise chalk catchments according to flow variability, average flows, the short-term response of flow to sharp bursts of rainfall, groundwater contributions to the flow regime, and (indirectly) the rate of recharge and runoff production. This helps inform which NFM interventions will be most suitable based on whether water transfer occurs predominantly above or below the ground surface. Details of index calculations are provided in Table 1.

### 2.2.2. Physical Catchment Properties

We selected 21 physical variables that directly influence hydrology and that have explained variations in catchment hydrology in other studies [29–31]. This included the percentage cover of 9 land uses (arable land, broad leaved and coniferous woodland, grassland, heathland, urban, suburban, marshland, and inland rock) from the 2015 Land Cover Map (LCM 2015) [32]. Heathland, grassland, and marsh land covers are made up of sub-categories of each classification, respectively (i.e., all separate grassland types are combined into a new, homogenous grassland classification). Combinations of sub-categories to generate the three new classes were done as specified in Appendix A of the LCM2015 documentation to generate more optimum data distributions. The hydrological impact of land cover is well documented in the literature (Table 1) and is relevant to NFM because land cover directly impacts flow pathways and, consequently, the choices of NFM interventions that may be suitable. Additionally, land use can inform the amount of space

available for NFM application to inform what is possible. Large quantities of storage ponds, controlled flood plain zones, and afforestation may not be suitable NFM interventions in regions of arable land due to the need to preserve economically productive space, whereas other interventions, like winter cover crops, hedge edges, and no-till farming, may be more suitable. Percentage cover of 6 soil types (bypass flow common, bypass flow uncommon, bypass flow very uncommon, bypass flow variable, slowly permeable, and impermeable) were classified according to the hydraulic conductivity in the Hydrology of Soil Types (HOST) [24]. This is because hydraulic conductivity can directly dictate the efficiency of aquifer recharge and surface-runoff generation. Soil-saturated hydraulic conductivity has been identified as one of the key features that dictates chalk stream river regime and flood response [31]. It is therefore important to understand the key hydrological processes in the catchment that will in turn be useful to inform the choice of NFM interventions. Topographic catchment shape and drainage density were included because they influence the rate of propagation of water through the catchment and river channel network [33,34] (p. 304). Bedrock transmissivity and a proxy for aquifer abstraction rates were included because they provide information about how water propagates through the sub surface of the catchment, which in turn influences streamflow response [35,36]. Transmissivity values of the chalk bedrock vary between 230 $m^2$/day to 2600 $m^2$/day [36]. Catchments where transmissivity is low have reduced rates of recharge, subsurface water transfer, and reduced groundwater contributions to channel flows relative to regions with high-transmissivity bedrock [36]. The hydrological influence of each physical catchment variable is included in Table 1 as well as details of their calculation.

**Table 1.** Equations and methodology of variables compiled for the response and explanatory databases for the redundancy analysis.

| Variable | | Description and Influence on Flow |
|---|---|---|
| Mean Discharge ($Q_{Mean}$) | $\frac{\sum Q}{n}$ $n$ = days in record | The average quantity of water in the river channel. Gives an indication of the discharge under normal conditions from the NRFA [26]. |
| Maximum Recorded Discharge ($Q_{Max}$) | $Q_{Max}$ | Maximum discharge capacity of catchments at the gauging site during high flows from the NRFA [26]. |
| Richard–Baker Flashiness Index (RBFI) | RBFI = $\frac{\sum_{i=1}^{n}\lvert q_i - q_{i-1}\rvert}{\sum_{i-1}^{n} q_i}$ $q$ = dimensionless measure of discharge $i$ = time $n$ = number of discharge measurements | Measures the absolute daily fluctuations in streamflow, divided by the sum of all stream flow for the time series length [37]. Values range between 0 and 1. Values near 0 represent stable flow and those close to 1 represent highly changeable flows and spate conditions [29]. |
| Base Flow Index | | BFI measures the proportion of river runoff derived from stored sources [25,26]. |
| Catchment Area (Ca) | - | Watershed boundary shapefiles from the NRFA [26] facilitate calculations for other catchment properties. Smaller catchments are associated with steeper hydrograph rising limbs due to reduced catchment complexity [38,39]. |
| Form Factor (Rf) | Rf = $\frac{Ca}{Bl^2}$ | Measures the geometric shape of the catchment. Circular catchments (Rf = 0) have steep hydrograph peaks [34] (p. 304). |
| Drainage Density (Dd) | Dd = $\frac{StL}{Ca}$ | High drainage densities are linked to drainage efficiency and high peak flows [33]. |
| Channel Slope | m/m | Calculated from a raster layer of SRTM 30m Digital Elevation Model [40] in ArcGIS using the zonal statistics function. |
| % Land Cover Type | $\frac{Landuse\ area}{Ca} \times 100$ | Land-use effects on hydrology [32]. |
| | Arable Land and Horticulture | Reduced peak flows [41]. |
| | Broadleaf Woodland | Reduced peak flows [42]. |
| | Coniferous Woodland | Reduced peak flows due to large leaf surface area [42]. |
| | Grassland | No overall influence [43]. |
| | Heathland | No overall influence—the effects of vegetation are counteracted by shallow soils and low storage capacity [44]. |
| | Urban | Increased peak flows—impervious surfaces and drainage systems [45]. |
| | Suburban | Increased peak flows—impervious surfaces and drainage systems [45]. |
| | Marshland | |
| | Inland Rock | Increased peak flows—imperviousness. |
| % Soil Cover Type | $\frac{HOST\ Soil\ Type}{Ca} \times 100$ | The Hydrology Of Soil Types (HOST) from CEH [24]. |
| | Bypass Flow Common | Permeable (Vertical saturated hydraulic conductivity > 10 cm/day$^{-1}$) |
| | Bypass Flow Uncommon | Permeable (Vertical saturated hydraulic conductivity > 10 cm/day$^{-1}$) |
| | Bypass Flow Very Uncommon | Permeable (Vertical saturated hydraulic conductivity > 10 cm/day$^{-1}$) |
| | Bypass Flow Variable | Semi-permeable (Vertical saturated hydraulic conductivity 0.1–10 cm/day$^{-1}$) |

**Table 1.** *Cont.*

| Variable | | Description and Influence on Flow |
|---|---|---|
| | Slowly Permeable Soils | Semi-Permeable (Vertical saturated hydraulic conductivity 0.1–10 cm/day$^{-1}$) |
| | Impermeable | Impermeable (<0.1 cm/day$^{-1}$ Vertical saturated hydraulic conductivity) |
| | Peat | Semi-permeable (Vertical saturated hydraulic conductivity 0.1–10 cm/day$^{-1}$) [46]. |
| Transmissivity (m$^2$/day) | Transmissivity $= \frac{ga^3}{12v}$ $g$ = acceleration due to gravity $a$ = area $v$ = kinematic viscosity of the fluid | The rate at which water passes through the chalk bedrock [35,36]. |
| Abstraction Score | Subjectively assigned an arbitrary abstraction score based on the flow regime description on the gauging station info in the NRFA. Scores: flow added; natural flow; minor, moderate and major reduction of flows due to abstraction. | Quantifies influence of abstraction on river regime [26]. |

### 2.3. Statistical Analysis

We used redundancy analysis (RDA) [47] to characterise and explain the variation in hydrological properties in relation to physical properties. RDA is an asymmetric ordination method whereby the variation in one set of (explanatory) variables is used to directly explain the variation in another set of (response) variables. Essentially a combination of multiple regression and principal component analysis, RDA generates a matrix of the fitted values of all response variables, which are then subjected to a principal component analysis. Linear combinations of explanatory variables (physical catchment properties) that best explain the variation in the response variables (hydrological properties) are sought by the model in successive order. The main advantages of using RDA over principal component analysis in this context are that variance in the response variables is attributed to the explanatory variables, and presence or absence of a relationship between specific x and y variables can be tested [47]. It can therefore be used to directly link hydrological traits of chalk streams to the presence or absence of specific physical features, enabling more targeted NFM scheme design.

All analyses were undertaken in the R environment (version 3.6.1) [48]. After compiling the variables as outlined in Table 1, all variables were transformed using a Box Cox transformation and were centred and scaled to reduce the influence of outliers and place variables on a common scale for the RDA model [49]. To identify the explanatory variables that best describe the co-variation of the hydrological variables, a preliminary redundancy analysis (RDA) model was initially run using the rda function of the vegan package in R [50]. This global model, using all 21 catchment variables, was subjected to model validation. To achieve a parsimonious model containing important physical explanatory variables, the global model was subjected to a forward stepwise procedure using the forward.sel function of the Packfor R package [51] (p. 48). This process selects the model with the combination of variables with the highest R$^2$ and *p* value [52] (p. 178). The remaining physical catchment variables after the stepwise procedure are those that directly correlate with and explain the variation in hydrological regime in chalk streams whilst maintaining the highest explanatory power (R$^2$). The resulting forward-selected RDA model was then subjected to a validation process, including ANOVA tests of the significance of the relationships identified in the parsimonious model and the number of significant axes in the model using 1000 permutations. The covariance of the selected physical catchment variables in the model was ascertained using the variance inflation factor (VIF). Catchments were then plotted as a points on a biplot according to their RDA coordinates.

Catchment Classification

Cluster analysis was used to group catchments with similar river regimes according to catchment RDA coordinates. Catchments with similar combinations of physical catchment property variables are located near each other on an RDA biplot (have similar RDA coor-

dinates). Before subjecting the data to cluster analysis, the data's tendency for clustering was established using the Hopkins Statistic with the get_clust_tendency function of the factoextra package in R [53]. Four hierarchical clustering methods were compared for their suitability to the dataset according to two distance measures (cophenetic correlation and the Gower statistic). Cophenetic correlation measures the degree of agreement between the original unmodeled pairwise distances and the pairwise distances in the dendrogram. A high positive correlation indicates that pairwise distances have been preserved [54]. The Gower statistic is the sum of squared differences between the dissimilarity matrix and the cophenetic distance and is calculated using dendrogram hierarchical partition [55,56]. The clustering algorithms compared were single, complete, and average linkage agglomerative clustering and Ward's minimum variance clustering [55]. The optimum number of clusters was established via a Mantel correlation. Here, the original distance matrix is compared to binary matrices computed from the dendrogram being cut at multiple different levels for different numbers of cluster allocations [55]. The optimum number of clusters is where the Mantel correlation is highest.

The uncertainty in the allocated clusters was estimated using silhouette widths. This measures the degree of membership of an object to its allocated cluster by comparing the average distance of an object to all other objects in the same cluster, to the average distance between it, and all the objects in the next closest cluster [57]. Accordingly, high silhouette width values indicate that that catchment has a high degree of membership in that group. Catchments with negative silhouette width values can be assumed to have been misclassified.

Once the clustering allocations were validated using the pairwise distance measures, the catchment groupings were mapped and used to describe the variation within chalk stream hydrology and flow pathways, and the potential for the application of NFM for each group was assessed according to its specific physical and hydrological qualities.

## 3. Results

The stepwise redundancy analysis revealed that the combinations of the following catchment properties significantly explained the co-variation of hydrological variables ($p \leq 0.001$): area, uncommon bypass flow, impermeable soils, slowly permeable soils, station elevation, form factor, and urban land use (Table 2). The adjusted $R^2$ values, representing the proportion of hydrological variance, which is explained by the physical catchment properties, were 0.682 for the global model and 0.675 for the reduced model. Only the first two axes were used for analysis (axes 1: 77.4%, axes 2: 35.9%), as the third axis and beyond were statistically insignificant. Variance inflation factor (VIF) values (Table 3) revealed that the remaining 9 variables have very little collinearity (VIF values < 5). Despite the fact that the variables of uncommon bypass flow and slowly permeable soils were mildly collinear, they were both retained because removal of either caused a substantial drop in the explanatory power of the model. The 15 catchment variables removed via the forward selection process did little to reduce the Adjusted $R^2$ value, suggesting that the discarded variables mostly generated noise.

**Table 2.** Explanatory variables chosen during the forward selection process along with their comparative explanatory power (adjusted R2) and significance (F statistic and *p* Values). The explanatory variables represent the most parsimonious model and explain the most variation in the hydrological response dataset.

| Variable | Order | $R^2$ | Cumulative $R^2$ | Adjusted Cumulative $R^2$ | F Statistic | *p* Value |
|---|---|---|---|---|---|---|
| Area | 2 | 0.357 | 0.357 | 0.353 | 108.6 | 0.001 |
| Uncommon Bypass Flow | 18 | 0.231 | 0.588 | 0.584 | 109.3 | 0.001 |
| Impermeable Soils | 22 | 0.034 | 0.622 | 0.584 | 17.5 | 0.001 |
| Slowly Permeable Soils | 21 | 0.037 | 0.659 | 0.652 | 20.8 | 0.001 |
| Station Elevation | 1 | 0.02 | 0.669 | 0.660 | 5.8 | 0.001 |
| Form Factor | 4 | 0.010 | 0.679 | 0.669 | 6.1 | 0.001 |
| Urban | 14 | 0.007 | 0.686 | 0.675 | 4.43 | 0.001 |

**Table 3.** VIF values of the explanatory variables kept in the parsimonious RDA model. VIF values exceeding 10 indicate collinearity between variables.

|  | Area | Uncommon Bypass Flow | Impermeable Soils | Slowly Permeable Soils |
|---|---|---|---|---|
| VIF | 1.143 | 4.625 | 1.691 | 4.482 |
|  | **Station Elevation** | **Form Factor** | **Urban** | |
| VIF | 1.248 | 1.029 | 1.057 | |

*Clustering and Cluster Validation*

Catchments were grouped into three clusters using the average linkage hierarchical clustering algorithm. The data's tendency for clustering was established prior to this via the Hopkins statistic (0.752; significant = >0.5 and <1). Average linkage hierarchical clustering was selected out of the four algorithms because it returned the highest cophenetic correlation and smallest value for the Gower distance (Table 4). Despite four clusters being identified as the optimum number of clusters for this dataset according to the Mantel correlation (Figure 3a), three clusters were used because the fourth was not associated with any vectors on the biplot, making it difficult to interpret. Silhouette widths were used to quantify and map the uncertainty in catchment group allocations (Figure 4c). An average silhouette width of 0.50 indicates that catchments were generally classified correctly, with all misclassifications occurring in group 1 (Figure 3b). Silhouette widths below 0.31 were considered uncertain due to being close to the decision boundary, and catchments with silhouette widths below 0 were misclassified [52] (p. 70). A drop in average silhouette width from 0.53 with four clusters to 0.5 with three clusters indicates that group cohesion is slightly reduced by this decision. To mitigate against this, the nine misclassified catchments (Figure 3b) were reclassified and displayed as their nearest neighbour alternative classification in Figure 4b.

**Table 4.** Gower distance and cophenetic correlation results for all four hierarchical clustering methods for comparison. A high cophenetic correlation indicates a strong relationship between the distance matrix and the cophenetic matrix describing the dendrogram split where each point is clustered differently. A lower Gower distance indicates an acceptable clustering algorithm.

|  | Single Linkage | Complete Linkage | Average Linkage | Ward's Clustering |
|---|---|---|---|---|
| Gower distance | 30,878.360 | 62,837.240 | | 6538.116 |
| Cophenetic correlation | 0.757 | 0.748 | | 0.802 |

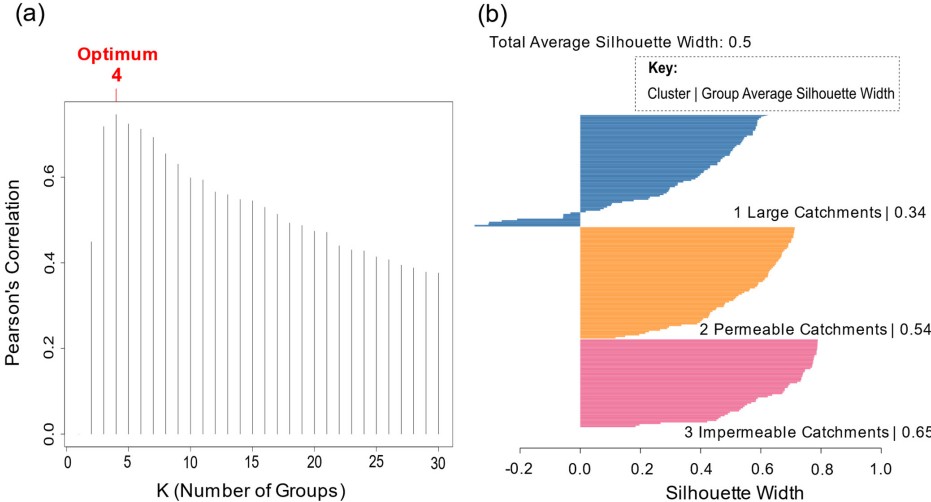

**Figure 3.** (**a**) The optimum number of average linkage clusters according to the Mantel statistic. The bars show the correlations between original distance matrix and binary matrices computed from the dendrogram cut at various levels. (**b**) Silhouette plot of the final three groups from the average linkage agglomerative clustering.

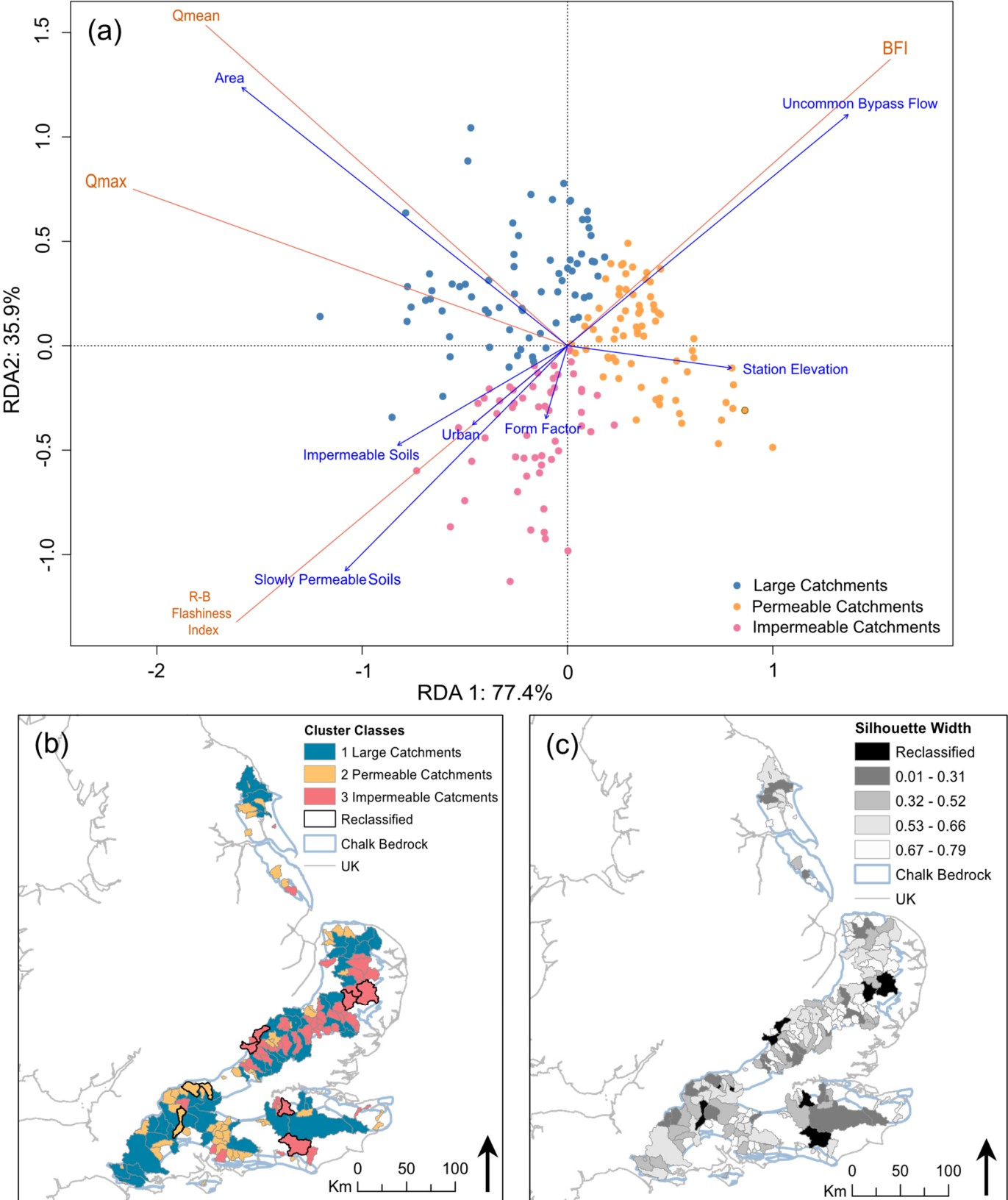

**Figure 4.** (**a**) Biplot showing catchment locations along two RDA axes. Dark orange vectors represent hydrological variables, and blue vectors are the catchment properties, with points for catchments placed relative to the vectors based on their individual values for catchment properties. (**b**) The catchment classifications mapped spatially. Misclassified catchments are presented in their reclassified grouping and outlined in bold. (**c**) The uncertainty of catchment classification according to silhouette widths. Misclassified catchments (<0) were reclassified into their nearest neighbour alternative classification.

## 4. Discussion

The explanatory power of the parsimonious RDA model is within the acceptable range for RDA models that characterise hydrological variation [58–60]. The model has been used to group catchments with similar physical catchment properties and river regime properties using their coordinates from the RDA plot. Thus, the cluster classifications were used to infer catchment typology and hydrological conditions, including rates of aquifer recharge and rapid runoff generation. The cluster classifications are used as a screening process to identify which NFM interventions can theoretically be applied or ruled out of each catchment typology based on the dominant physical catchment properties identified by that group as shown on the RDA plot (Figure 4).

It is acknowledged that this approach relies heavily on the partitioning of catchments into groups. Hence, care was taken to ensure that uncertainty in cluster allocation was reduced by checking the data's tendency to cluster. The number of clusters and the clustering algorithm that made the most statistical sense were selected. However, uncertainty in cluster allocation cannot be removed entirely, and 13.6% of the sample had uncertain group allocations (Figure 4c). It is understood that these catchments share traits that would allow them to be comfortably grouped in to two of the three groups and are therefore understood to be intermediate catchments that do not easily fit in to a single classification. In these cases, the combined traits of the two catchment classifications should be taken into account when selecting NFM options for these catchments (alternative groupings for highly uncertain catchments are provided in Appendix A). Catchments that have been reclassified can be confidently allocated to their group.

### 4.1. Group 1: Large Catchments

Group 1 catchments have high $Q_{mean}$ and $Q_{max}$ values that are explained by their large size (Figure 4). These are the largest catchments within the sample, with topographic catchment areas ranging from 108 km$^2$ to 1459 km$^2$. $Q_{max}$ has a strong negative correlation to station elevation, demonstrating that catchments with higher discharges tend to be those closer to sea level and further downstream, resulting in large water accumulation. $Q_{mean}$, $Q_{max}$, and catchment area have weak correlations (shown by orthogonality) with all other variables in the model (Figure 4a), rendering it difficult to comment further.

Previous research suggests that NFM becomes less effective at reducing flooding in catchments greater than 20 km$^2$ [11,61]. Furthermore, evidence shows that increasing the area impacted by NFM measures does not always increase the gains in flow attenuation [13,41]. The lack of substantial evidence for NFM implementation at a larger catchment scale has become a barrier to the general uptake of NFM [62]; however, it must be acknowledged that very few catchment-scale NFM schemes have been implemented [13,41]. The NFM evidence base at this scale is mostly provided by risk-based predictive models [63], which introduces computation limitations to catchment-scale NFM research and their subsequent findings [64]. Nonetheless, it is generally accepted that NFM is most effective for (and possibly limited to) catchments < 20 km$^2$.

Metcalfe et al. (2018) [64] argued that a large proportion of the benefits of NFM come from its ability to de-synchronise sub-catchment flood waves, which theoretically works at every catchment scale [65]. De-synchronisation is achieved by implementing NFM schemes at a small scale in carefully chosen sub-catchments that are designed to attenuate individual sub-catchment flood waves, reducing the likelihood of flood wave synchronisation. Whilst it is recognised that the flood peak reductions from this process are often small, it can be enough to prevent bank overtopping in events up to 1:100 year scale [66]. Additionally, this method is only suitable in situations where the configuration of the sub-catchments currently causes synchronised flood waves. Dixon et al. (2016) [67] illustrate that, in some cases, NFM placement can synchronise flood waves that were previously staggered, resulting in an overall increase in flood risk. Where desynchronisation of flood waves is the goal, experimental modelling is required in the design process and prior to implementation of an NFM scheme to mitigate such risks [64,68].

As none of the catchments found in group 1 are <20 km$^2$, the evidence is lacking to support the effective use of NFM within these river basins. However, this does not prevent these larger catchments from being broken down into multiple sub-catchments or for NFM opportunities to be identified and applied locally for point-source flooding. An example of this could be to implement bunds on a sloped field to prevent field runoff from flooding an adjacent road. In specific situations, it may be suitable to investigate the use of de-synchronising flood waves to reduce the incidence of bank overtopping. Such areas would be downstream river sections where multiple sub-catchments contribute to flood waves, and relatively small changes in river levels alter the risk of bank overtopping [64,66]. Extensive research and considerable time are required to design and implement effective NFM strategies for de-synchronisation [68], so it is only recommended where combined flood waves are known to be a problem. Low drainage density of permeable chalk catchments, however, limits the opportunities for de-synchronisation.

### 4.2. Group 2: Permeable Catchments

These catchments are the stereotypical chalk streams: small catchments that are dominated by groundwater recharge and influxes of groundwater that dominate the river regime via baseflow. Group 2 catchments have river regime variability closely related to high BFI values and are associated with a large proportion of uncommon bypass flow soil types and higher station elevations (interpreted here as a higher proportion of headwater catchments). Uncommon bypass flow is a classification of soils from the HOST soil classification that describes thin (aquifer within 2m), permeable, and unconsolidated soils with micro and macro pores [24]. These soils have a vertical saturated hydraulic conductivity of >10 cm day$^{-1}$ [69], allowing highly efficient recharge via vertical drainage into the chalk aquifer. This is linked to reduced runoff generation [70], limiting the potential for NFM implementation because a large proportion of the water transfer throughout the catchment occurs below the ground surface. Due to the lack of significant runoff, NFM treatments that aim to reduce runoff generation or store surface water within the catchment are highly unlikely to have a significant effect on streamflow. Examples of such interventions include winter cover crops, changes in tillage practices, lengthening drainage pathways, planting across slopes, online and offline storage ponds, wetlands, and controlled flood zones [13]. What remains are NFM techniques that focus on in-channel interventions to reduce downstream conveyance [13], such as channel realignment, sustainable urban drainage systems (SuDS), de-culverting covered river channels, and increasing in-channel, riparian, and marginal vegetation [6,71]. Whilst in-channel river restoration schemes such as these can significantly reduce flood peaks in small catchments [8,72], it must be emphasised that restoration schemes and in-channel interventions deliver the best results as part of a suite of other NFM measures [73]. Therefore, whilst beneficial, in-channel measures alone are unlikely to deliver the optimum impact of NFM interventions.

Despite a lack of surface-runoff generation in highly groundwater-dominated catchments, such as those in group 2, surface water can occur due to groundwater emergence. In this case, the water table rises to intersect with the ground surface, forming static pools of water in topographic low points called turloughs [74] and intermittently flowing river channels called winterbournes [75] and can activate springs in weak points and fractures in the chalk [20,76]. These phenomena generally occur after large quantities of prolonged rainfall. Previous groundwater-emergence floods have been recorded after rainfall events that double and triple the long-term averages [74,77–79]. Under such conditions, NFM has been shown to be far less effective because engineered and natural stores of water, such as soil storage, storage ponds, log jams, and groundwater stores, become full and are overwhelmed [11,13]. The presence of turloughs, winterbournes, and springs demonstrate that water transfer throughout the catchment is not uniform across space and is dictated by the location, size, and activation of hydrogeological features such as fractures [19,80–82]. As a result, NFM schemes in these catchments will require in-depth, local knowledge to design them, with sensitivity to the local hydrogeological features. We suggest that NFM

interventions in this group of catchments be focused in areas where groundwater emergence occurs as winterbournes and springs to intercept or store flood flows, particularly where the resulting surface water causes disruption or damage to property. For example, in-channel interventions, such as log jams and online storage ponds, could be installed in in the path of known winterbournes and springs to intercept flows when the water table is high. The potential benefits of this kind of NFM installation have not currently been tested.

For these catchments, it is suggested that NFM will have diminished effectiveness due to a lack of significant surface runoff. This limits suitable NFM interventions to mostly instream channel modifications to reduce rapid conveyance, which will be less effective compared to a full suite of NFM interventions [13,73]. In many cases, it may be more economical to map and maintain local knowledge of groundwater emergence for flood-risk mapping. Previously, major flood damages were caused by urbanisation of forgotten, dormant winterbournes and springs, which then activate under heavy rainfall [78]. These mapping efforts would also be instrumental in designing and locating potential NFM schemes near hydrogeological features, such as springs and winterbournes. Therefore, effective flood mitigation measures are associated with tracking water table heights, flood warnings, flood mapping, and making a concerted effort to understand spatial changes in hydrogeology. Long-term groundwater emergence and flood-risk maps are required for appropriate flood planning, for identifying locations for NFM schemes, and for reducing flood risk by restricting building and development on areas at risk of groundwater emergence.

### 4.3. Group 3: Less Permeable Catchments

Group 3 catchments are associated with high RBFI values (Figure 4), which is best explained by higher incidence of impermeable and slowly permeable soils, the presence of urban land use, and higher values of form factor (indicating more circular catchment shapes). These are the chalk catchments with the largest proportions of surface runoff. Higher RBFI values describe greater flow variability and steeper and higher magnitude discharge peaks. The relationship between soil permeability and rapid runoff generation is well known and documented, where impermeable soils and surfaces generate greater quantities of surface runoff and quick-flow catchment pathways [24,31,70,83,84]. Work previously performed on chalk stream catchments by Ascott et al. (2017) [31] concluded that impermeable superficial deposits, such as those found in group 3 catchments, slow the vertical conveyance of water into the aquifers, reducing recharge and groundwater dominance in the river regime and flood response. Inversely, reduced rate of recharge and absorption of rainfall sub-surface will increase surface-runoff generation.

As demonstrated by Lane (2017) [13], a large proportion of NFM interventions work by manipulating and intercepting surface-runoff pathways [61]. By virtue of a greater quantity of surface runoff, the full suite of NFM strategies are viable in group 3 catchments, including reduction of rapid runoff generation through soil management and increased catchment roughness, increasing catchment water storage using storage ponds, and reducing the conveyance of water downstream with in-channel and river restoration strategies. This allows many different potential combinations of NFM intervention for optimising results [73], meaning that this group of chalk catchments is the most suitable for the application of NFM in the study sample.

### 4.4. Applications of NFM in Chalk Catchments

According to the typology of catchments generated via redundancy analysis, three chalk catchments in the UK (Yeading Brook West at North Hillingdon, Catchwater at Withernwick, and the River Dour at Crabble Mill; Appendix A) inherently have the physical and hydrological features best suited to the current range of NFM measures. The study sample accounts for 198 of the estimated 224 chalk streams in the UK. In the study sample, only 25 chalk catchments are <20 km$^2$, and of these, 22 are classified as permeable catchments, limiting them to mostly in-channel NFM interventions and highly targeted NFM schemes

downstream of hydrogeological features. Overall, this suggests that implementing NFM in chalk groundwater-dominated catchments is likely to have sub-optimal results compared to other catchment types [3,12,85].

The findings of this analysis do not necessarily negate the use of NFM methods in chalk catchments. Where appropriate, de-synchronising sub-catchment flood waves can be implemented for larger catchments classed as impermeable by applying NFM at the local scale. NFM should also be considered on the local scale in areas where groundwater emergence as springs and winterbournes cause disruptions or for any other known source of surface water or runoff. Additionally, the environmental benefits of NFM on water quality, aesthetic improvements, reductions in soil erosion, and biodiversity are uncontested [62]. There are proven benefits for river managers and catchment partnerships in chalk streams regions to implement NFM for river restoration, water quality, and biodiversity improvements. The application of NFM is a cross-disciplinary, collaborative process, so the benefits of increased communication and the formation of catchment partnerships have been argued to be another of the co-benefits [61]. Wingfield et al. (2019) [62] argued that building the evidence base for NFM to irrefutability could take decades and that by hesitating to implement NFM due to a lack of current evidence, potential benefits are lost. Minor flood benefits gained through river restoration will likely have minimal effect for large-scale storms but may reduce the incidence of small-scale nuisance floods [66,67]. Therefore, river restoration is advantageous to flood reduction but should not be considered the main objective unless supported by local, detailed analysis. Additionally, even in groups 1 and 2 catchments, there are likely small-scale point-source locations of pluvial flooding, like flooding of fields or roads, and groundwater emergence that could be combatted with small-scale NFM schemes. However, implementation of such schemes will require better reporting and mapping of local sources of flooding.

## 5. Conclusions

The results of a redundancy analysis model were used to generate a typology of chalk catchments, resulting in three groupings according to broadly similar river regimes and physical catchment properties. Using these classifications as a screening tool, the likely effectiveness of applying NFM in each of these groups was discussed. The first class (group 1) is grouped by virtue of their larger size, meaning they are likely unsuited to NFM due to NFM being most effective for catchments <20 km$^2$. It is acknowledged that these catchments can be broken down into smaller sub-catchments, possibly to de-synchronise sub-catchment flood waves. De-synchronisation is only recommended in conjunction with hydrological modelling prior to NFM design. It should also be considered that flood reductions from de-synchronising flood waves are most effective for smaller nuisance floods rather than larger flood events. Permeable catchments (group 2) are associated with smaller headwater catchments and high-permeability soils. NFM interventions are less suited due to a low proportion of surface-runoff processes. Large quantities of surface water can, however, be generated due to groundwater emergence at hydrogeological features, such as winterbournes or springs, which activate when the water table is high. Effective flood planning in these catchments is more likely to come in the form of in-channel NFM interventions, hydrogeological mapping, building, and planning restrictions and the development of groundwater emergence early warning systems. We suggest that NFM schemes in these catchments be small-scale and highly targeted to deal with runoff from activated hydrogeological features that would otherwise cause small-scale disruptions (i.e., flooding roads). Catchments classed as impermeable (group 3) are related to the presence of impermeable and slowly permeable soils as well as other less-permeable surfaces, such as urban land use. Due to this, runoff is generated, meaning that this category of catchments is the most suited to NFM in the chalk stream sample.

Overall, this study suggests that implementing NFM in chalk groundwater-dominated catchments is likely to have sub-optimal results compared to other catchment types. However, NFM implementation may be justifiable purely on the merit of the multiple environ-

mental benefits, such as improved water quality, aesthetic improvements, reduced soil erosion, increased biodiversity, and collaboration across multiple river-management bodies. This paper provides a first order triage of the potential for NFM runoff-management methods in chalk catchments. Further work in this field will need to focus on hydrological models that represent the permeability of the soils and the influence of groundwater on stream flows, detailed groundwater emergence, and flood-risk mapping as well as NFM-implementation studies specifically on chalk streams.

**Author Contributions:** I.B., D.S., J.S. and J.L. conceived and designed the study. I.B. collated the data and undertook the analysis, with statistical modelling input from R.S. I.B. drafted the manuscript, which was then edited by R.S., J.S., J.L., T.S. and D.S. All authors have read and agreed to the published version of the manuscript.

**Funding:** This research was funded by the UK Environment Agency.

**Institutional Review Board Statement:** Not applicable.

**Informed Consent Statement:** Not applicable.

**Data Availability Statement:** The raw data and the R code script generated for and used for the analysis in this study can be found on Figshare. 10.6084/m9.figshare.14200229.

**Acknowledgments:** We thank the Environment Agency and the School of Geography and Environmental Science, University of Southampton, for support to Imogen Barnsley. Thanks to Katharine Smith for her editorial assistance.

**Conflicts of Interest:** The authors declare no conflict of interest.

## Appendix A. Catchments in Their Groups after Redundancy Analysis

| | | | Group 1—Large Catchments | | |
|---|---|---|---|---|---|
| No. | Station Number | River | Location | Catchment Area (km²) | Alternative Group Where Classification Is Uncertain |
| 1 | 26001 | West Beck | Wansford Bridge | 195.56 | |
| 2 | 26002 | Hull | Hempholme Lock | 397.23 | |
| 3 | 26004 | Gypsey Race | Bridlington | 257.15 | |
| 4 | 26005 | Gypsey Race | Boynton | 248.13 | 2 |
| 5 | 26009 | West Beck | Snakeholme Lock | 195.61 | |
| 6 | 27087 | Derwent | Low Marishes | 475.93 | |
| 7 | 33003 | Cam | Bossington | 807.32 | |
| 8 | 33004 | Lark | Isleham | 464.16 | |
| 9 | 33006 | Wissey | Northwold Total | 259.36 | |
| 10 | 33007 | Nar | Marham | 147.39 | |
| 11 | 33013 | Sapiston | Rectory Bridge | 196.16 | |
| 12 | 33014 | Lark | Temple | 274.04 | |
| 13 | 33016 | Cam | Jesus Lock | 769.19 | |
| 14 | 33019 | Thet | Melford Bridge | 311.38 | |
| 15 | 33021 | Rhee | Burnt Mill | 308.05 | |
| 16 | 33022 | Ivel | Blunham | 539.63 | |
| 17 | 33023 | Lea Brook | Beck Bridge | 131.89 | 3 |
| 18 | 33024 | Cam | Dernford | 199.59 | |
| 19 | 33028 | Flit | Shefford | 119.42 | |
| 20 | 33034 | Little Ouse | Abbey Heath | 707.76 | |
| 21 | 33044 | Thet | Bridgham | 274.99 | |
| 22 | 33057 | Ouzel | Leighton Buzzrad | 122.40 | |
| 23 | 34003 | Bure | Ingworth | 161.27 | |
| 24 | 34004 | Wensum | Costessey Mill | 559.70 | |
| 25 | 34006 | Waveney | Needham Mill | 376.08 | |

| | | | Group 1—Large Catchments | | |
|---|---|---|---|---|---|
| No. | Station Number | River | Location | Catchment Area (km²) | Alternative Group Where Classification Is Uncertain |
| 26 | 34011 | Wensum | Fakenham | 162.93 | 2 |
| 27 | 34014 | Wensum | Swantom Morely Total | 377.46 | |
| 28 | 34019 | Bure | Horstead Mill | 327.91 | |
| 29 | 36001 | Stour | Stratford St Mary | 838.01 | |
| 30 | 36006 | Stour | Langham | 571.35 | |
| 31 | 36015 | Stour | Lamarsh | 481.28 | 3 |
| 32 | 38001 | Lee | Feildes Weir | 1045.10 | |
| 33 | 38018 | Upper Lee | Water Hall | 157.73 | 3 |
| 34 | 38031 | Lee | Rye Bridge | 758.52 | |
| 35 | 39003 | Wandle | South Wimbledon | 153.73 | |
| 36 | 39010 | Colne | Denham | 725.90 | |
| 37 | 39013 | Colne | Berrygrove | 349.25 | 3 |
| 38 | 39016 | Kennet | Theale | 1037.87 | |
| 39 | 39019 | Lambourn | Shaw | 235.43 | |
| 40 | 39023 | Wye | Hedsor | 134.18 | 2 |
| 41 | 39027 | Pang | Pangbourne | 175.68 | |
| 42 | 39030 | Gade | Croxley Green | 182.35 | |
| 43 | 39031 | Lambourn | Welford | 158.87 | |
| 44 | 39043 | Kennet | Knighton | 299.17 | 2 |
| 45 | 39078 | Wey (North) | Farnham | 192.60 | 2 |
| 46 | 39103 | Kennet | Newbury | 534.13 | |
| 47 | 39104 | Mole | Esher | 471.36 | |
| 48 | 39115 | Pang | Bucklebury | 108.94 | |
| 49 | 40003 | Medway | Teston | 1261.311 | |
| 50 | 40008 | Great Stour | Wye | 226.40 | 3 |
| 51 | 40011 | Great Stour | Horton | 341.27 | |
| 52 | 40012 | Darent | Hawley | 187.31 | |
| 53 | 40016 | Cray | Crayford | 123.49 | 2 |
| 54 | 40018 | Darent | Lullington | 122.24 | 2 |
| 55 | 41004 | Ouse | Barcombe Mills | 400.61 | |
| 56 | 41009 | Rother | Hardham | 360.77 | |
| 57 | 42010 | Itchen | Highbridge & Allbrook Total | 339.91 | |
| 58 | 42012 | Anton | Fullerton | 186.16 | 2 |
| 59 | 42016 | Itchen | Easton | 234.17 | 2 |
| 60 | 42024 | Test | Chilbolton Total | 478.43 | |
| 61 | 43003 | Avon | East Mills Total | 1459.44 | |
| 62 | 43004 | Bourne | Laverstock | 165.21 | |
| 63 | 43005 | Avon | Amesbury | 326.47 | |
| 64 | 43006 | Nadder | Wilton | 215.63 | |
| 65 | 43007 | Stour | Throop | 1062.09 | |
| 66 | 43008 | Wylye | South Newton | 448.17 | |
| 67 | 43018 | Allen | Walford Mill | 170.82 | 2 |
| 68 | 43024 | Wylye | Stockton Park | 170.89 | 2 |
| 69 | 44001 | Frome | East Stoke Total | 414.59 | |
| 70 | 44002 | Piddle | Baggs Mill | 183.80 | |
| 71 | 44004 | Frome | Dorchester Total | 205.67 | |

| | Group 2—Catchments with More Permeable Soils | | | | |
|---|---|---|---|---|---|
| No. | Station Number | River | Location | Catchment Area (km²) | Alternative Group Where Classification Is Uncertain |
| 1 | 26003 | Foston Beck | Foston Mill | 59.59 | |
| 2 | 26006 | Elmswell Beck | Little Driffield | 133.18 | 1 |
| 3 | 26008 | Mires Beck | North Cave | 40.96 | |
| 4 | 26013 | Driffield Trout | Driffield | 53.34 | |
| 5 | 26016 | Gypsey Race | Kirby Grindalythe | 16.98 | |
| 6 | 27073 | Brompton Beck | Snainton Ings | 8.06 | |
| 7 | 29001 | Waithe Beck | Brigsley | 108.75 | |
| 8 | 29003 | Lud | Louth | 56.42 | 3 |
| 9 | 33025 | Babingly | West Newton Mill | 44.69 | |
| 10 | 33032 | Heacham | Heacham | 56.16 | |
| 11 | 33033 | Hiz | Arlesey | 112.96 | |
| 12 | 33040 | Rhee | Ashwell | 2.00 | |
| 13 | 33049 | Stanford Water | Buckenham Tofts | 46.45 | |
| 14 | 33052 | Swaffham Lode | Swaffham Bulbeck | 33.12 | |
| 15 | 33054 | Babingley | Castle Rising | 48.54 | |
| 16 | 33056 | Quy Water | Lode | 92.56 | |
| 17 | 33061 | Shep | Fowlmere One | 1.03 | |
| 18 | 33062 | Guilden Brook | Fowlmere Two | 3.40 | |
| 19 | 33064 | Whaddon Brook | Whaddon | 14.53 | |
| 20 | 33065 | Hiz | Hitchin | 12.03 | |
| 21 | 33068 | Cheney Water | Gatley End | 0.11 | |
| 22 | 34012 | Burn | Burnham Overy | 83.87 | |
| 23 | 34018 | Stiffkey | Warham | 86.13 | 3 |
| 24 | 38017 | Mimram | Whitwell | 38.40 | |
| 25 | 39015 | Whitewater | Lodge Farm | 46.99 | |
| 26 | 39029 | Tilling Bourne | Shalford | 58.78 | |
| 27 | 39032 | Lambourn | East Shefford | 145.06 | 1 |
| 28 | 39033 | Winterbourne Stream | Bagnor | 45.31 | |
| 29 | 39036 | Law Brook | Albury | 16.07 | |
| 30 | 39037 | Kennet | Marlborough | 136.43 | 1 |
| 31 | 39039 | Wye | High Wycombe | 67.73 | |
| 32 | 39061 | Letcombe Brook | Letcombe Bassett | 3.99 | |
| 33 | 39065 | Ewelme Brook | Ewelme | 11.98 | |
| 34 | 39077 | Og | Marlborough Poulton Farm | 63.95 | |
| 35 | 39091 | Misbourne | Quarrendon Mill | 65.87 | |
| 36 | 39101 | Aldbourne | Ramsbury | 53.09 | |
| 37 | 39102 | Misbourne | Denham Lodge | 93.24 | |
| 38 | 39107 | Hogsmill | Ewell | 8.44 | |
| 39 | 39112 | Letcombe Brook | Arabellas Lake | 3.10 | |
| 40 | 39113 | Manor Farm Brook | Letcombe Regis | 1.38 | |
| 41 | 39114 | Pang | Frilsham | 90.06 | |
| 42 | 39118 | Wey | Alton | 44.50 | |
| 43 | 39119 | Wey | Kings Pond (Alton) | 46.15 | |
| 44 | 39120 | Caker Stream | Alton | 83.94 | |
| 45 | 39146 | Mill Brook | Blewbury | 2.01 | |
| 46 | 39147 | Wendover Springs | Wendover | 9.49 | |
| 47 | 40013 | Darent | Otford | 98.19 | |
| 48 | 40014 | Wingham | Durlock | 30.72 | |

| | | Group 2—Catchments with More Permeable Soils | | | |
|---|---|---|---|---|---|
| No. | Station Number | River | Location | Catchment Area (km²) | Alternative Group Where Classification Is Uncertain |
| 49 | 40033 | Dour | Crabble Mill | 44.93 | |
| 50 | 41015 | Ems | Westbourne | 57.89 | |
| 51 | 41023 | Lavant | Graylingwell | 86.20 | |
| 52 | 41033 | Costers Brook | Cocking | 2.74 | |
| 53 | 41034 | Ems | Walderton | 42.46 | |
| 54 | 41037 | Winterbourne Stream | Lewes | 17.48 | |
| 55 | 42005 | Wallop Brook | Broughton | 53.46 | |
| 56 | 42006 | Meon | Mislingford | 75.84 | |
| 57 | 42007 | Alre | Drove Lane Alresford | 57.45 | |
| 58 | 42008 | Cheriton Stream | Sewards Bridge | 74.33 | |
| 59 | 42009 | Candover Stream | Borough Bridge | 72.06 | |
| 60 | 42015 | Dever | Weston Colley | 50.15 | |
| 61 | 42025 | Lavant Stream | Leigh Park | 55.96 | |
| 62 | 42026 | Wallop Brook | Bossington | 61.13 | |
| 63 | 42027 | Dever | Bransbury | 122.36 | 1 |
| 64 | 43010 | Allen | Loverley Farm | 94.84 | |
| 65 | 43011 | Ebble | Bodenham | 105.55 | |
| 66 | 43012 | Wylye | Norton Bavant | 114.01 | 1 |
| 67 | 43014 | East Avon | Upavon | 85.82 | |
| 68 | 44006 | Sydling Water | Sydling St Nicholas | 12.05 | |
| 69 | 44008 | South Winterbourne | Winterbourne Steepleton | 20.18 | 3 |
| 70 | 44009 | Wey | Broadwey | 8.00 | |
| 71 | 101003 | Lukely Brook | Carisbrooke Mill | 14.86 | |
| | | Group 3—Catchments with Less Permeable Soils | | | |
| No. | Station Number | River | Location | Catchment Area (km²) | Alternative Group Where Classification Is Uncertain |
| 1 | 26007 | Catchwater | Withernwick | 10.84 | |
| 2 | 29002 | Great Eau | Claythorpe Mill | 80.42 | |
| 3 | 33011 | Little Ouse | County Bridge Euston | 129.34 | |
| 4 | 33027 | Rhee | Wimpole | 128.49 | |
| 5 | 33029 | Stringside | Whitebridge | 95.41 | |
| 6 | 33030 | Clipstone Brook | Clipstone | 40.35 | |
| 7 | 33045 | Wittle | Quidenham | 27.45 | |
| 8 | 33046 | Thet | Redbridge | 14.43 | |
| 9 | 33050 | Snail | Fordham | 57.88 | |
| 10 | 33051 | Cam | Chesterford | 140.02 | |
| 11 | 33053 | Granta | Stapleford | 113.98 | |
| 12 | 33055 | Granta | Babraham | 101.97 | |
| 13 | 33066 | Granta | Linton | 61.61 | |
| 14 | 33070 | Lark | Fornham St Martin | 111.07 | |
| 15 | 34001 | Yare | Colney | 228.81 | |
| 16 | 34002 | Tas | Shotesham | 153.19 | |
| 17 | 34005 | Tud | Costessey Park | 72.11 | |
| 18 | 36002 | Glem | Glemsford | 85.61 | |
| 19 | 36004 | Chad Brook | Long Melford | 50.33 | |
| 20 | 36005 | Brett | Hadleigh | 155.85 | |
| 21 | 36007 | Belchamp Brook | Bardfield Bridge | 58.16 | |
| 22 | 36008 | Stour | Westmill | 222.82 | |
| 23 | 36009 | Brett | Cockfield | 25.56 | |

| | | | Group 3—Catchments with Less Permeable Soils | | |
|---|---|---|---|---|---|
| No. | Station Number | River | Location | Catchment Area (km²) | Alternative Group Where Classification Is Uncertain |
| 24 | 36010 | Bumpstead Brook | Broad Green | 27.58 | |
| 25 | 36011 | Stour Brook | Sturmer | 34.24 | |
| 26 | 36012 | Stour | Kedington | 76.65 | |
| 27 | 36013 | Brett | Higham | 191.78 | |
| 28 | 37012 | Colne | Poolstreet | 64.49 | |
| 29 | 37016 | Pant | Copford | 63.80 | |
| 30 | 38002 | Ash | Mardock | 78.00 | |
| 31 | 38003 | Mimram | Panshanger Park | 130.21 | |
| 32 | 38004 | Rib | Wadesmill | 136.79 | |
| 33 | 38005 | Ash | Easneye | 84.65 | |
| 34 | 38006 | Rib | Herts Training School | 148.67 | |
| 35 | 38011 | Mimram | Fulling Mill | 99.25 | |
| 36 | 38012 | Stevenage Brook | Bragbury Brook | 35.11 | |
| 37 | 38013 | Upper Lee | Luton Hoo | 70.31 | |
| 38 | 38028 | Stanstead Brook | Gypsy Lane | 26.37 | 2 |
| 39 | 38029 | Quin | Griggs Bridge | 50.41 | |
| 40 | 38030 | Beane | Hartham | 173.85 | |
| 41 | 39004 | Wandle | Beddington Park | 117.35 | 2 |
| 42 | 39005 | Beverly Brook | Wimbledon Common | 39.49 | |
| 43 | 39012 | Hogsmill | Kingston Upon Thames | 72.91 | |
| 44 | 39014 | Ver | Hansteads | 134.55 | |
| 45 | 39028 | Dun | Hungerford | 100.09 | |
| 46 | 39055 | Yeading Brook West | North Hillingdon | 1.31 | |
| 47 | 39088 | Chess | Rickmansworth | 96.91 | |
| 48 | 39089 | Gade | Bury Mill | 44.73 | |
| 49 | 39125 | Ver | Redbourn | 62.58 | |
| 50 | 39127 | Misbourne | Little Missenden | 47.26 | 2 |
| 51 | 40015 | White Drain | Fairbrook Farm | 31.40 | |
| 52 | 40027 | Sarre Pen | Calcott | 19.59 | |
| 53 | 41003 | Cuckmere | Sherman Bridge | 130.33 | |
| 54 | 41028 | Chess Stream | Chess Bridge | 25.01 | |
| 55 | 42001 | Wallington | North Fareham | 111.68 | |
| 56 | 42011 | Hamble | Frogmill | 55.25 | |

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
