# Peer review of "Exploring the Capability of Natural Flood Management Approaches in Groundwater-Dominated Chalk Streams"

_water, doi:10.3390/w13162212_

Round 1
Reviewer 1 Report
This study assessed the capability of NFM approaches in groundwater-dominated chalk streams over the 198 catchments in the UK using the redundancy analysis method. Their results suggest that implementing NFM in chalk groundwater-dominated catchments is likely to have sub-optimal results compared to other catchment types. Overall, the paper is well written and is within the scope of the journal. Therefore, I recommend it to be accepted for publication after some minor revisions.
- L197, is it table 1?
- Section 2.3, more detailed introductions for ‘Redundancy Analysis’ are needed.
- Figure 4(b) is difficult to read. Please consider using brighter colors.
- ‘Catchment classification’ is key point of this study. How does the uncertainty in the allocated clusters affect the result of three groups of catchments? Related discussion may be needed.
- I suggest the authors add ‘catchment area’ in Appendix A.
Reviewer 2 Report
Methods of the research are explained very generally.
- Table 2 (195) should be Table 1!
- Transmissivity (m3/day) ??? should be m2/day!
- It is not necessary to present the appendix (groups of catchments), I think it has no sense for the reader.
Round 2
Reviewer 2 Report
My comments have been included in the text. Mistakes have been improved.